# Modulatory Effect of *Chelidonium majus* Extract and Its Alkaloids on LPS-Stimulated Cytokine Secretion in Human Neutrophils

**DOI:** 10.3390/molecules25040842

**Published:** 2020-02-14

**Authors:** Sylwia Zielińska, Monika Ewa Czerwińska, Magdalena Dziągwa-Becker, Andrzej Dryś, Mariusz Kucharski, Anna Jezierska-Domaradzka, Bartosz J. Płachno, Adam Matkowski

**Affiliations:** 1Department of Pharmaceutical Biology, Wroclaw Medical University, Borowska 211, 50-556 Wroclaw, Poland; anna.jezierska-domaradzka@umed.wroc.pl (A.J.-D.); pharmaceutical.biology@wp.eu (A.M.); 2Department of Pharmacognosy and Molecular Basis of Phytotherapy, Medical University of Warsaw, Banacha 1, 02-097 Warsaw, Poland; monika.czerwinska@wum.edu.pl; 3Department of Weed Science and Tillage Systems, Institute of Soil Science and Plant Cultivation State Research Institute, Orzechowa 61, 50-540 Wrocław, Poland; m.dziagwa@iung.wroclaw.pl (M.D.-B.); m.kucharski@iung.wroclaw.pl (M.K.); 4Department of Physical Chemistry, Wroclaw Medical University, Borowska 211A, 50-556 Wroclaw, Poland; andrzej.drys@umed.wroc.pl; 5Department of Plant Cytology and Embryology, Jagiellonian University, Gronostajowa 9, 30-387 Kraków, Poland; bartosz.plachno@uj.edu.pl; 6Laboratory of Experimental Cultivation, Botanical Garden of Medicinal Plants, Wroclaw Medical University, Al. Jana Kochanowskiego 14, 50-556 Wroclaw, Poland

**Keywords:** isoquinoline alkaloids, cytokine secretion, cytotoxicity, greater celandine

## Abstract

Due to certain differences in terms of molecular structure, isoquinoline alkaloids from *Chelidonium majus* engage in various biological activities. Apart from their well-documented antimicrobial potential, some phenanthridine and protoberberine derivatives as well as *C. majus* extract present with anti-inflammatory and cytotoxic effects. In this study, the LC–MS/MS method was used to determine alkaloids, phenolic acids, carboxylic acids, and hydroxybenzoic acids. We investigated five individually tested alkaloids (coptisine, berberine, chelidonine, chelerythrine, and sanguinarine) as well as *C. majus* root extract for their effect on the secretion of IL-1*β*, IL-8, and TNF-*α* in human polymorphonuclear leukocytes (neutrophils). Berberine, chelidonine, and chelerythrine significantly decreased the secretion of TNF-*α* in a concentration-dependent manner. Sanguinarine was found to be the most potent inhibitor of IL-1*β* secretion. However, the overproduction of IL-8 and TNF-*α* and a high cytotoxicity for these compounds were observed. Coptisine was highly cytotoxic and slightly decreased the secretion of the studied cytokines. The extract (1.25–12.5 μg/mL) increased cytokine secretion in a concentration-dependent manner, but an increase in cytotoxicity was also noted. The alkaloids were active at very low concentrations (0.625–2.5 μM), but their potential cytotoxic effects, except for chelidonine and chelerythrine, should not be ignored.

## 1. Introduction

Isoquinoline alkaloids contained in *Chelidonium majus* L. extracts have been found to be some of the most potent molecules against bacteria, viruses, fungi, and protozoa [1]. This group of compounds is represented mostly by phenanthridine, protoberberine, and protopine derivatives. Roots are richer in these compounds [2,3], but the *European Pharmacopoeia* [4] only refers to the alkaloid content of herbs (which should not be less than 0.6% and is calculated in terms of chelidonine content). According to the available literature, roots contain up to 4% alkaloids, while the aerial parts contain 0.1–1% (with the exception of coptisine, which is much more abundant, especially in fruits). Due to differences in their molecular structures, *C. majus* alkaloids engage in different biological activities [1]. Quaternary nitrogen in molecules, e.g., in chelerythrine, sanguinarine, berberine, or coptisine, has been found to be involved in the inhibition of cellular respiration, whereas the presence of an imine moiety in the structures of sanguinarine and chelerythrine determines the ability of these compounds to inhibit the activity of proteins and enzymes [5].

Abundant data have been gathered since the first experiments on the anti-inflammatory activity of *C. majus*. Due to the low toxicity of *C. majus* and its antimicrobial and anti-inflammatory activity, it is recommended for the treatment of oral inflammatory conditions [6]. Over the years, many reports have appeared on the high cytotoxicity of *C. majus* extracts and individual compounds: these have also pertained to their specific mechanisms of action [7,8,9,10,11,12,13,14].

Apart from structure, which has a decisive impact on the profile and strength of biological activity, the presence of other compounds in plant extracts is also of great importance. In studies independently examining pro- and anti-inflammatory properties, isoquinoline alkaloids have appeared to be effective agents that are highly active even at concentrations 50 times lower than, for instance, polyphenols. In the present study, we examined *C. majus* root extract and five individual alkaloids for their effects on cytokine secretion in human neutrophils as well as their cytotoxic potential. The phytochemical profile of the *C. majus* root extract was performed using liquid chromatography with a triple-quadrupole analyzer (LC–MS/MS).

## 2. Results

### 2.1. Phytochemical Analysis

In the phytochemical profile of the *C. majus* root extract, a total of 25 compounds were identified in negative (12 compounds) or positive (13 compounds) electrospray ionization mode. Seven isoquinoline alkaloids—two protopine derivatives (protopine, allocryptopine), two protoberberine derivatives (coptisine, berberine), and three phenanthridine derivatives (chelidonine, sanguinarine, chelerythrine)—as well as unidentified derivatives of protopine (**12**), chelidonine (**18**), tetrahydroberberine (**21**), tetrahydrocoptisine (**22**), and coptisine (**23**) were detected (Table 1). The protopine derivative (**12**) exhibited the most abundant precursor ion at *m*/*z* 354, and after disintegration, the product ion at *m*/*z* 320.2 corresponded to protopine (**25**, *R*_t_ = 12.76 min). For allocryptopine (**13**), a parent ion at *m*/*z* 369.6 was chosen, as were product ions at *m*/*z* 352, 187.95, and 290. The assignment of the chelidonine derivative (**18**, *R*_t_ = 8.84 min) was based on the characteristic loss of a fragment of *m*/*z* 353.8, which was attributable to a protonated molecule of chelidonine (**19**, *R*_t_ = 9.56 min). For tetrahydroberberine (**21**), also known as canadine, the most abundant ion was a precursor ion at *m*/*z* 340, while for berberine (**16**), it was a parent ion at *m*/*z* 336.4 and product ions at *m*/*z* 320, 292, and 321.1. Coptisine (**15**) produced a parent ion at *m*/*z* 320.1, whereas tetrahydrocoptisine (**22**) and one other coptisine derivative (**23**) produced a parent ion at *m*/*z* 324. Chelerythrine (**20**, *R*_t_ = 10.44 min) and sanguinarine (**24**, *R*_t_ = 11.80 min) showed parent ions at *m*/*z* 348.1 and 332.1, respectively.

A quantitative analysis revealed that no individual compound predominated in the extract, although large differences between alkaloids and compounds from other chemical groups were observed in the contents (Table 1). Next to sanguinarine (1373.80 µg/g), chelerythrine, which was the most abundant (1761.22 µg/g), chelidonine (1181.41 µg/g), and coptisine (1077.06 µg/g) were detected in relatively large proportions (Figure 1). Derivatives of protopine, chelidonine, coptisine, and berberine were also present, but in amounts below LOQ (Table 1). Three nonaromatic carboxylic acids (malic acid (**1**, dicarboxylic acid), *trans*-aconitic acid (**2**, polycarboxylic acid), and quinic acid (**3**, cyclohexanecarboxylic acid)); three hydroxybenzoic acids (salicylic acid (**7**), protocatechuic acid (**4**), and vanillic acid (**5**)); two hydroxycinnamic acids (*trans*-caffeic acid (**6**) and *p*-coumaric acid (**9**)); one caffeic acid ester (rosmarinic acid (**10**)); one flavonoid (quercetin (**17**)); one phenolic aldehyde (vanillin (**11**)); and one quinine sulfate (**14**) were detected in the root extract.

### 2.2. Bioactivity Assays

Berberine, chelidonine, and chelerythrine decreased the secretion of TNF-*α* in human neutrophils in a concentration-dependent manner. The cells treated with chelidonine and chelerythrine at concentrations of 0.625 μM released 46.4% ± 12.4% and 83.1% ± 10.5% of TNF-*α*, respectively, compared to (+) LPS-treated cells (100% ± 9.8%). It is worth noting that neither chelidonine nor chelerythrine were toxic in this cell model (Figure 2). Only berberine at the highest concentration of 2.5 µM significantly increased the number of dead cells. However, berberine at a concentration of 0.625 μM inhibited TNF-*α* (secretion of 60.1% ± 11.4% compared to (+) LPS, which was 100% ± 9.8%), and it was not toxic at this concentration.

Sanguinarine was the most potent inhibitor of IL-1*β* secretion. However, the overproduction of IL-8 and TNF-*α* was observed at lower concentrations (Figure 3B, Figure 4B, Figure 5B, and Figure 6A–D). This compound, at 2.5 μM, significantly inhibited both IL-8 and TNF-*α* release (by neutrophils) compared to other concentration levels, but it was also highly cytotoxic. However, the secretion of IL-1*β* by cells treated with sanguinarine at a concentration of 0.625 μM, which was not cytotoxic, was reduced to 52.2% ± 3.1% compared to the negative control LPS-stimulated cells.

Coptisine slightly decreased the secretion of cytokines tested in this study, but it was highly cytotoxic in a concentration range of 0.625–2.5 μM (Figure 2B, Figure 3B, Figure 4B, Figure 5B and Figure 6A–D).

*C. majus* root extract induced the secretion of all cytokines in a dose-dependent manner in a concentration range from 1.25 µg/mL to 12.5 µg/mL (Figure 3A, Figure 4A, Figure 5A). It was highly cytotoxic at concentrations of 6.25 and 12.5 µg/mL (Figure 2A) as well as at a higher concentration range of 25–100 µg/mL (data not shown). It is worth noting that the extract, at a concentration of 1.25 µg/mL, did not exert any cytotoxic effects compared to nontreated cells (Figure 2A), but its influence on cytokine release was insignificant (Figure 3A, Figure 4A, Figure 5A).

## 3. Discussion

The extract from the *C. majus* roots collected for this study (in early spring (9th of April)) was rich in chelerythrine, sanguinarine, chelidonine, and coptisine (Table 1), without the significant predominance of any of these alkaloids. Sanguinarine was not the most abundant compound, as has been previously reported for roots harvested later in the vegetation season (end of May and June) [3] and in autumn (data not published). Coptisine has been previously found to be the most abundant alkaloid in the aerial parts of plants, particularly in fruits [3]. However, the roots are rather poor in this alkaloid. Additionally, a much larger amount of berberine in root extract was found in the current study, whereas the compound was nearly absent in both aerial and underground parts collected later in the vegetation season, as well as in organ and tissue in vitro cultures [2,3]. Not much is known about isoquinoline alkaloid conversion along the biosynthesis pathway. A biosynthetic pathway for different classes of isoquinoline alkaloids has been proposed for *Coptis japonica* (Thunb.) Makino and *Papaver somniferum* L [15]. The authors examined the production stability of benzylisoquinoline alkaloids in *C. japonica*-transformed plants for 4 and 20 months. Chromatographic analyses revealed that plants cultivated for a longer period accumulated alkaloids at higher levels [15]. The mechanisms involved in alkaloid transport across the tonoplast are also not very well understood. Berberine is one of the best known isoquinoline alkaloids in this respect. A suggested model for berberine transport and accumulation in the vacuoles of *Coptis japonica* cells has been presented by Otani et al. [16]. In an intact tonoplast and vacuole vesicle system, berberine uptake was stimulated by Mg/pyrophosphate, Mg/adenosine triphosphate (ATP), guanosine-5′-triphosphate (GTP), cytidine triphosphate (CTP), and uridine-5′-triphosphate (UTP). On the other hand, ammonium cation and bafilomycin A1 strongly inhibited berberine uptake, whereas vanadate, which is used to inhibit ATP-binding cassette transporters, presented only a slight effect, which suggested the occurrence of a typical secondary transport mechanism [16].

The share of individual alkaloids in latex composition influences the biological activity profile of raw materials. In the present study, coptisine and sanguinarine were found to be responsible for the highest cytotoxicity, and they were produced in relatively high proportions in roots (1077.06 and 1373.80 µg/g of dry weight, respectively) (Figure 2, Table 1), which could explain a similar effect in the tested root extract. Sanguinarine and *C. majus* root extract have previously been found to present with cytotoxic and proapoptotic activity in studies on several cell lines obtained from patients with various types of leukemia [7]. In the same study, a similar effect was also shown by berberine, but it was much weaker than sanguinarine. The authors tested only these two alkaloids, and thus their interpretation of the extract’s effects was limited to these substances. In our experiments, chelidonine and chelerythrine proved to be the least cytotoxic (Figure 2 and Figure 6). A previously tested chelidonine-rich extract also showed similarly weak cytotoxic effect in in vitro studies on HepG2 cells treated with extracts prepared using different solvents. In turn, extracts containing higher amounts of coptisine and sanguinarine showed much stronger properties of this type [17]. To date, other experiments have shown that mitochondrial toxicity is associated with the ability of chelerythrine and sanguinarine to intercalate DNA [18]. Due to the cholagogue and hepatoprotective activities of *C. majus*, the main applications for this plant pertain to biliary tract and liver disorders, something recently reviewed by Zielińska et al. [1]. Nonetheless, existing clinical evidence is not sufficient to recommend extracts of the herb for usage (European Medicines Agency, 2011) [19]. In previous studies on its mechanisms of action, chelidonine was found to present with cytostatic activity, whereas berberine (**16**), chelerythrine, and sanguinarine presented with DNA-intercalating abilities, thereby interfering with cell division and replication [10,11,13,14,20]. It is worth noting that the application of *Chelidonium*-derived products in the treatment of cancer has also been suggested [21]. On the other hand, *Chelidonii herba* has been traditionally used in the treatment of gastrointestinal disorders, including gall bladder dysfunctions, as a choleretic and cholagogue agent [1]. Its traditional use has been cited in studies on acute cholestatic hepatitis [22]. Thus, researching the anti-inflammatory activity of *C. majus* constituents, which may cause the inhibition of both infectious and noninfectious inflammation related to chronic inflammatory diseases, seems to be justified. Neutrophils constitute the first line of defense against pathogens, due to their ability to generate oxidative bursts and release antimicrobial substances. Lipopolysaccharide, a component of the outer membranes of Gram-negative bacteria, elicits a cytokine-mediated innate immune response, and therefore it was used as a cell stimulator in our study [23].

The qualitative analysis revealed that phenolic acids, hydroxybenzoic acids, carboxylic acids, flavonoids, and several of their derivatives were present in the root extract, in addition to alkaloids (Table 1). These components of *C. majus* root extract may have an important impact on its bioactivity profile (Figure 2, Figure 3, Figure 4, Figure 5 and Figure 6). Even though the inhibitory activity of cytokine secretion by selected alkaloids has been observed, the relevant activity of the root extract has not been noted. Therefore, compounds other than the alkaloids detected in the extract are likely to attenuate the inhibitory effect of alkaloids in human neutrophils. Phenolic compounds, as well as some other constituents, have been previously detected in the roots and aerial parts of this species [24,25,26,27,28], and research on the anti-inflammatory properties of, e.g., polyphenolic compounds [29] has shown that their activity occurs at relatively high concentrations. The modulatory effect on LPS-stimulated cytokine secretion in human neutrophils was influenced not only by the nature of the chemical compounds but also by their potency (Figure 2, Figure 3, Figure 4, Figure 5 and Figure 6). Our studies show that the isoquinoline alkaloids were much more potent, as they were active at over 50-times-lower concentrations (Figure 3, Figure 4 and Figure 5) than were the other groups of compounds, such as the polyphenols [29,30]. In the anti-inflammatory test conducted by Granica et al. [29], simple bibenzyls, benzylphthalides, and cannabispiradienone; simple phenolic acids; and flavonoids from *Tragopogon tommasinii* Sch. Bip. were active at concentrations of 500 µM. In a study by Czerwińska et al. [30], flavonoids, phenylpropanoid glycosides, and iridoids were active at concentrations of 5 and 25 µM.

Alkaloids found in *C. majus* root extract represent three classes—protopine derivatives, protoberberine derivatives, and phenanthridine derivatives—although in this case, the phenanthrene skeleton as well as the presence of quaternary nitrogen atoms was less important. Both chelerythrine and sanguinarine, as well as berberine and coptisine, contain quaternary nitrogen in their structure, and yet these compounds present with different bioactivity effects, which was also shown in our current study. Similarly, the phenanthrene skeleton present in the structure of chelidonine, chelerythrine, and sanguinarine did not make these three compounds act similarly in terms of cytokine secretion. In previous studies by other researchers, the phenanthrene skeleton had no significant impact on oxygen uptake at the cellular level, whereas other building elements, such as the charge of the molecule, were crucial [5].

Despite the similar effects observed on TNF-*α* secretion and the relatively low cytotoxicity, berberine, chelidonine, and chelerythrine affected IL-1*β* and IL-8 secretion differently. The strongest TNF-*α* inhibition was observed in the case of nontoxic chelidonine. On the other hand, the significant inhibition of IL-8 and TNF-*α* secretion by sanguinarine at a concentration of 2.5 μM was probably caused by increased toxicity at this concentration. In addition, sanguinarine significantly inhibited IL-1*β* release by PMNs, even at a nontoxic concentration. The complex role of cytokines in the inflammatory pathways should be underlined. Firstly, the results of the study pointed to multiple mechanisms that may be engaged in the release of interleukin-1*β* within the same population of cells. The mechanisms of IL-1*β* secretion depend on factors such as the cell microenvironment, temperature, pH, and osmolarity. These conditions change when inflammatory processes escalate and more mechanisms are involved. In addition, more and more endogenous molecules that are released from dead cells, such as danger-associated molecular patterns (DAMPs), can appear in a changing cell microenvironment, leading to the depletion of the inflammatory response. Therefore, the close relationship between cytotoxic effects and cytokine secretion might explain the different results in terms of the inhibition of cytokine secretion when the cells were treated with alkaloids [31]. However, it should also be noted that even though DAMPs were generated, the isoquinoline alkaloids were able to limit noninfectious inflammation and reduce the cell-to-cell spreading of inflammation. A second important issue is that the major role of cytokines such as IL-8 can be seen as an induction of the chemotaxis of granulocytes in response to infection or tissue injury. IL-8 can stimulate the resolution of infection through the induction of phagocytosis and oxidative burst or the induction of cell proliferation and activation of an angiogenic response. Intereleukin-8 expression has been recognized in numerous cancer types. Tumor-derived IL-8 suppresses antitumor immunity through neutrophil recruitment. Thus, the concomitant IL-8 inhibitory effect and the toxicity of some of the tested alkaloids, such as sanguinarine, may reflect potential antitumor effects [32].

## 4. Material and Methods

### 4.1. Plant Material

*Chelidonium majus* L. root and root collars were collected on 9 April 2019 in Poland (Wrocław, Okólna Street, 51°06′02.0″ N 17°03′34.0″ E51.100545, 17.059446). Dried raw material (70 g) was powdered using mortar and pestle. The extracts were prepared in round-bottom flasks in an ultrasonic bath (2 × 2 h), with a solvent-to-solid ratio of 1:20 (*v*/*w*). Methanol and 50 mM hydrochloric acid were used for sample extraction, in accordance with procedures used in previous studies [33,34].

### 4.2. Phytochemical Analysis

The identification and quantification of the *C. majus* extract was performed using liquid chromatography–electrospray ionization–tandem mass spectrometry (LC–ESI–MS/MS) (Shimadzu, Kyoto, Japan).

#### 4.2.1. Reagents and Reference Substances

Water, LC–MS-grade methanol, and eluent-additive LC–MS ultra-ammonium formate (NH_4_HCO_2_) were purchased from Fluka Analytical (St. Louis, MO, USA).

Alkaloid standards such as chelidonine (purity ≥ 95%), sanguinarine (purity ≥ 90%), chelerythrine (purity ≥ 90%), and berberine (purity ≥ 95%) were purchased from Extrasynthese (France). Coptisine (purity ≥ 98%), p-coumaric acid, tannic acid, salicylic acid, chlorogenic acid, vanillic acid, *trans*-caffeic acid, vanillin, rosmarinic acid, quinine sulfate, and quercetin were purchased from Sigma (St. Louis, MO, USA).

Water was deionized and purified by ULTRAPURE Millipore Direct-QVR 3UV-R (Merck, Darmstadt, Germany).

#### 4.2.2. Liquid Chromatography

A liquid chromatography (LC) analysis was carried out with the use of a Shimadzu Prominence UFLC system (Shimadzu, Kyoto, Japan). It was equipped with a binary solvent manager (LC-30 ADXR), a degasser (DGU-20A3), a column oven (CTO-10ASVP), an autosampler (SIL 20AXR), and a system controller (CBM-20A), and it was interfaced to a triple-quadrupole analyzer. The chromatographic separation was performed using a Kinetex column (C18, 2.6 μm, 100 × 3.0 mm, Phenomenex, Torrance, CA, USA). The column temperature was maintained at 35 °C. The mobile phase was prepared as a mixture (A, B) consisting of 10 mM ammonium formate in water (A) and 0.1% formic acid in methanol (B). The methanol percentage was changed linearly: 0 min, 10%; 10 min, 85%; 13.01 min, 85%; 16 min, 10%. A sample volume of 10 µL was injected into the UHPLC system.

#### 4.2.3. Mass Spectrometry

The analysis was performed using a triple-quadrupole tandem mass spectrometer (LCMS-8030, Shimadzu, Kyoto, Japan) with ultrafast polarity switching and ultrafast multiple reaction monitoring (MRM) transitions. Nebulizing gas (obtained from pressurized air in an N2 LC–MS pump) and nitrogen were dried, working at a flow rate of 3 L/min and 15 L/min, respectively. The desolvation line temperature and the heat block temperature were 250 °C and 400 °C, respectively. Argon 99% (Linde, Wroclaw, Poland) was used as a collision-induced dissociation gas (CID) at a pressure of 230 kPa. A dwell time of 10 ms was selected. To process the quantitative data, LabSolution Ver. 5.6 (Shimadzu, Kyoto, Japan) software was used.

#### 4.2.4. Identification and Quantification

A full scan as well as MS/MS spectra were obtained during flow injection analysis (FIA) of each standard. For the identification and quantification of analytes, multiple reaction monitoring (MRM) mode was used. For most of the compounds, at least two transitions were chosen: the first one for quantitative purposes (Q) and the second one as confirmation (q). This was accomplished with a retention time and ion intensity ratio compared to corresponding reference substances and compounds (reported previously) [2].

The limit of detection (LOD) was calculated according to a signal-to-noise ratio (S/N) of 3, and the limit of quantitation (LOQ) was calculated according to an S/N ratio of 10. Linearity was evaluated from the square correlation coefficients (r^2^) of the regression curves obtained for each standard, and r^2^ ≥ 0.99 was achieved for all compounds. The changes in retention time, expressed as relative standard deviation, were not significant. The RSD% for the retention time never exceeded 2.5%.

### 4.3. Bioactivity Assays

#### 4.3.1. Chemicals

L-glutamine, HEPES buffer solution, fetal bovine serum (FBS), and RPMI 1640 medium were purchased from Sigma-Aldrich (St. Louis, MO, USA). Lipopolysaccharide (LPS from *Escherichia coli* 0111:B4) was purchased from Merck. Human ELISA sets (interleukin 1 beta (IL-1*β*), interleukin 8 (IL-8), tumor necrosis factor alpha (TNF-*α*), and propidium iodide (PI)) were purchased from BD Biosciences (New Jersey, U.S.). Phosphate-buffered saline (PBS) was obtained from Gibco. Penicillin and streptomycin were purchased from PAA (Cölbe, Germany). The water used in the study was purified using a water purification system (MILLIPORE Simfilter Simplicity UV, Merck, Darmstadt, Germany).

#### 4.3.2. Neutrophil Polymorphonuclear Granulocyte (PMN) Isolation

Buffy coats were obtained from the Warsaw Blood Donation Center (Warsaw, Poland). Donors of the blood (<35 years old) were diagnosed as healthy, according to their medical history and a routine laboratory test. Donors declared that they were not taking medication and that they were nonsmokers. Their good health condition was clinically confirmed, and a routine laboratory test revealed values within a normal range. Polymorphonuclear neutrophils (PMNs) were isolated using dextran sedimentation and centrifugation in a Ficoll Hypaque gradient (1500 rpm, 4 °C). Hypotonic lysis was performed to remove erythrocytes. The obtained cells were suspended in an RPMI 1640 medium [35].

#### 4.3.3. Cytotoxicity

Neutrophil cytotoxicity was assessed by flow cytometry with PI staining. The *C. majus* root extract was studied in a concentration range from 1.25 to 12.5 μg/mL, whereas the compounds were tested at concentrations of 0.625 µM, 1.25 µM, and 2.5 µM. After 20 h, the neutrophils (2 × 10^6^/mL in RPMI 1640 medium) were incubated with the tested *C. majus* root extract and the test compounds. Cell centrifugation was performed at 2000 rpm for 10 min at 4 °C. Neutrophils were rinsed twice with PBS, resuspended in PBS (500 μL) containing 5 μL of PI (50 μg/mL), and then left for 15 min in the dark. The cells were analyzed using flow cytometry with a FACSCalibur (Becton Dickinson, Franklin Lakes, NJ, USA), and the data was recorded (10,000 events). These cells, which displayed high permeability to propidium iodide, were expressed as a percentage of PI (+) cells. As a positive control, TritonX was used. The results of the viability were calculated according to an equation: 100% − PI (+) cells%.

#### 4.3.4. IL-8, TNF-α, and IL-1β Production by PMNs

PMNs (2 × 10^6^/mL) were maintained in 96-well plates in RPMI 1640 medium and with 10% FBS, 2 mM L-glutamine, 10 mM HEPES, and penicillin and streptomycin (1%, *v*/*v*) for 20 h at 37 °C with 5% CO_2_, and in the absence or presence of the tested compounds. The compounds were added 30 min before stimulation with lipopolysaccharide (100 ng/mL; LPS). The *C. majus* root extract was studied in a concentration range from 1.25 to 12.5 μg/mL, whereas the compounds were tested at concentrations of 0.625 µM, 1.25 µM, and 2.5 µM. Dexamethasone (Dex) was used as a positive control.

The TNF-*α*, IL-8, and IL-1*β* released into cell supernatants were measured using an enzyme-linked immunosorbent assay (ELISA) according to the instructions of the manufacturer. The effect on IL-8, TNF-*α*, and IL-1*β* production was calculated as a percentage of the released agent compared to the stimulated control (+) LPS (without the extract being tested).

### 4.4. Statistical Evaluations

Statistical significance showing differences between means was established using an ANOVA followed by a Brown–Forsythe test. Here, *p*-values below 0.05 were considered statistically significant. The results are expressed as the mean ± SE. The first step was to check which values were considered to be outliers by the statistical program. Then, the normality of the distribution was checked, and an analysis of variance was used. Additionally, a Kruskal–Wallis test (nonparametric) was performed. To examine the influence of the type of alkaloid and the alkaloid concentration on cytokine secretion or cytotoxicity, a two-way ANOVA was performed (Figure 6A–D). All analyses were conducted using Statistica ver. 10 (StatSoft, Cracow, Poland).

## 5. Conclusions

*C. majus* extract and its alkaloids exhibited strong modulatory effects on the three cytokines secreted. The results of our research indicate that coptisine and sanguinarine had the largest effect in terms of the mechanisms associated with the development of a cytotoxic effect in LPS-stimulated human neutrophils. On the other hand, sanguinarine was found to be one of the most potent inhibitors of IL-1*β* secretion, which was not reflected in the activity of the plant extract. Isoquinoline alkaloids were also found to be active at very low concentrations. In further experiments with selected substances based on the results obtained in the current study, the range of concentrations of the tested substances will have to be expanded in order to determine the threshold concentration at which a cytotoxic effect is observed. Moreover, future experiments based on testing mixtures of selected alkaloids in various proportions relative to each other would allow us to explore mechanisms of possible synergy (the addition or abolition of biological effects).

## Figures and Tables

**Figure 1 molecules-25-00842-f001:**
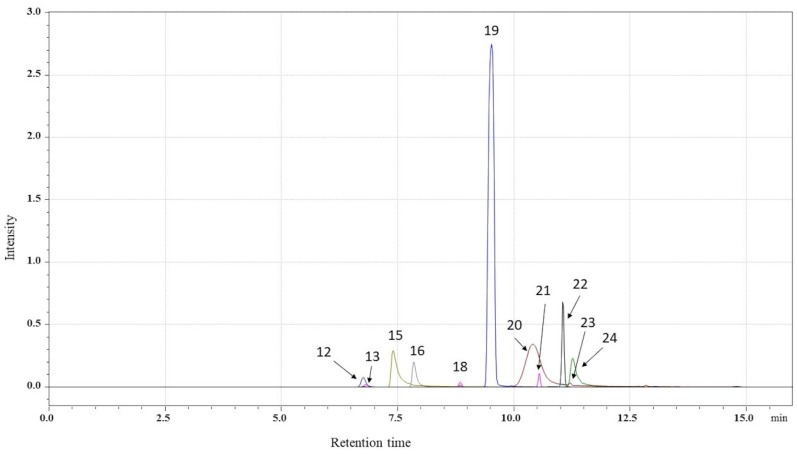
Multiple reaction monitoring (MRM) chromatogram of alkaloids in the crude acidified MeOH extract of *C. majus* roots.

**Figure 2 molecules-25-00842-f002:**
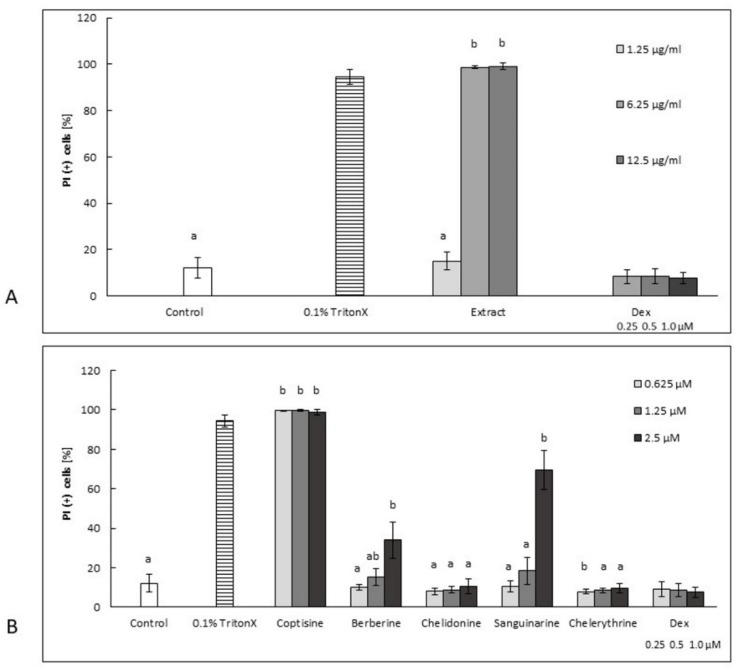
The cytotoxic effects of root extract (**A**: 1.25–12.5 µg/mL) and alkaloids (**B**: 0.625–2.5 µM) on neutrophils expressed as a percentage of PI (+) cells. Control, (+) LPS. Dex-dexamethasone at concentrations of 0.25, 0.5, and 1.0 µM. The statistical significance of the compounds and extract versus the stimulated control at *p* < 0.05 is marked by two different letters: a and b (ANOVA followed by a Kruskal–Wallis test).

**Figure 3 molecules-25-00842-f003:**
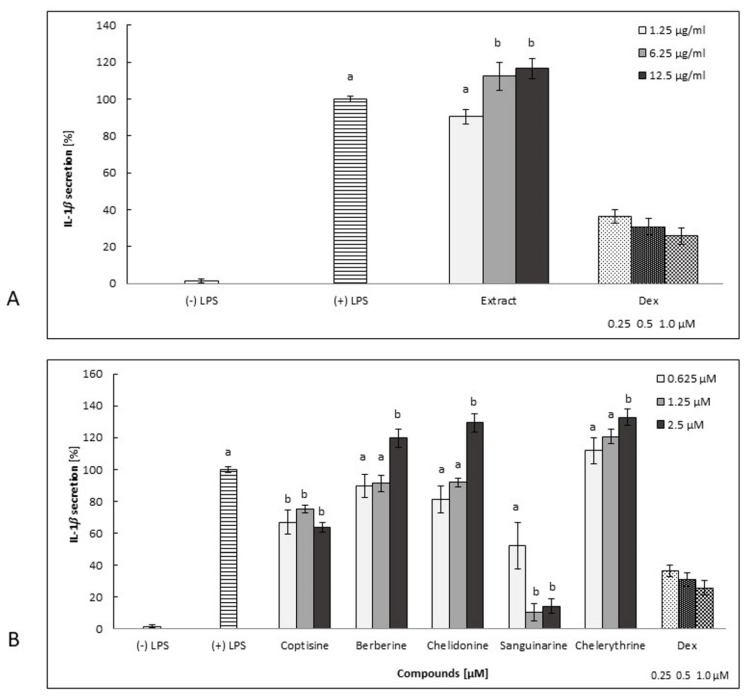
The effect of root extract (**A**: 1.25–12.5 µg/mL) and alkaloids (**B**: 0.625–2.5 µM) on neutrophil IL-1*β* release by LPS-stimulated (100 ng/mL) cells ((+) LPS). Dex-dexamethasone at concentrations of 0.25, 0.5, and 1.0 µM. The data are expressed as the mean ± SE from three donors assayed in triplicate. The statistical significance of the compounds and extract versus the stimulated control at *p* < 0.05 is marked by two different letters: a and b (ANOVA followed by a Kruskal–Wallis test).

**Figure 4 molecules-25-00842-f004:**
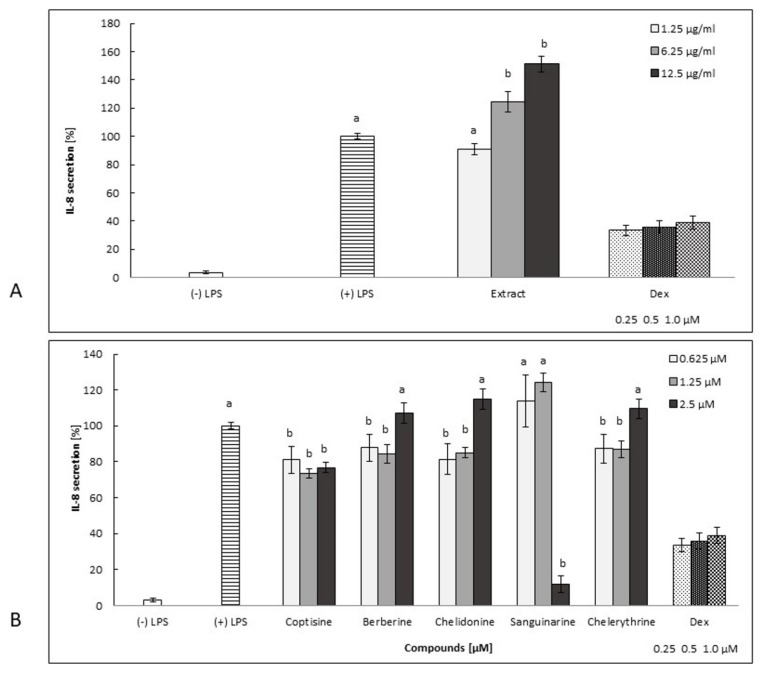
The effect of root extract (**A**: 1.25–12.5 µg/mL) and alkaloids (**B**: 0.625–2.5 µM) on neutrophil IL-8 release by LPS -stimulated (100 ng/mL) cells ((+) LPS). Dex-dexamethasone at concentrations of 0.25, 0.5, and 1.0 µM. The data are expressed as the mean ± SE from three donors assayed in triplicate. The statistical significance of the compounds and extract versus the stimulated control at *p* < 0.05 is marked by two different letters: a and b (ANOVA followed by a Kruskal–Wallis test).

**Figure 5 molecules-25-00842-f005:**
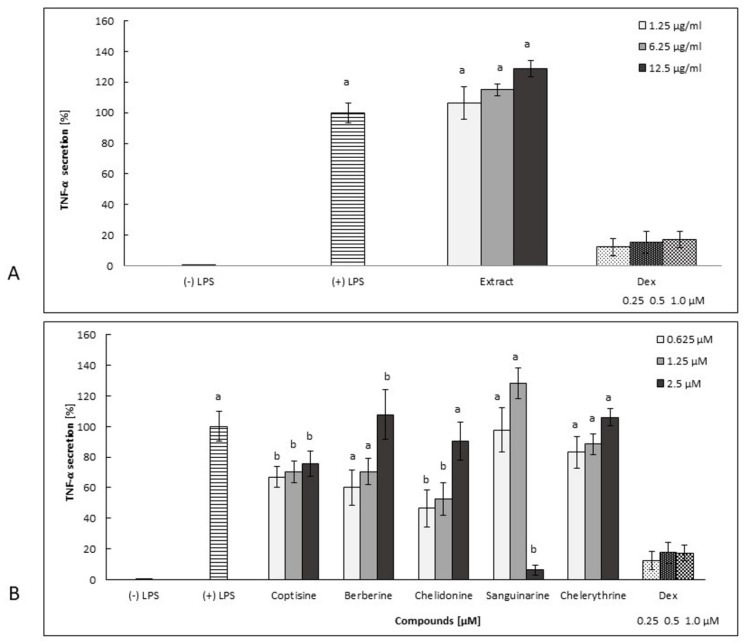
The effect of root extract (**A**: 1.25–12.5 µg/mL) and alkaloids (**B**: 0.625–2.5 µM) on neutrophil TNF-*α* release by LPS -stimulated (100 ng/mL) cells ((+) LPS). Dex-dexamethasone at concentrations of 0.25, 0.5, and 1.0 µM. The data are expressed as the mean ± SE from three donors assayed in triplicate. The statistical significance of the compounds and extract versus the stimulated control at *p* < 0.05 is marked by two different letters: a and b (ANOVA followed by a Kruskal–Wallis test).

**Figure 6 molecules-25-00842-f006:**
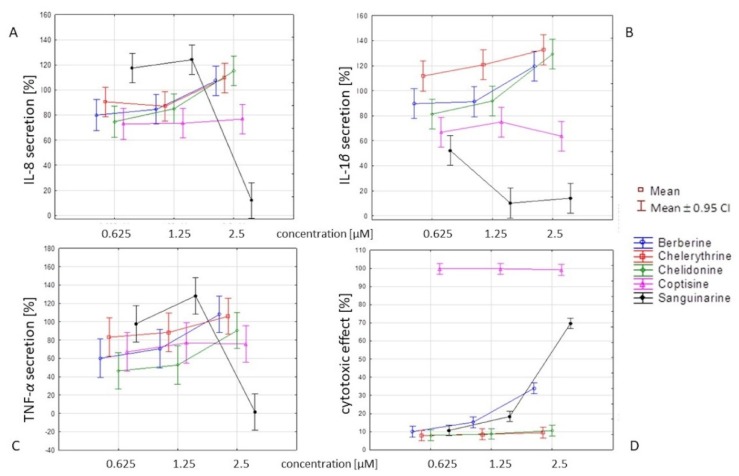
The results of a two-way ANOVA by type and concentration of the alkaloid in terms of the secretion of three cytokines (**A**–**C**) and cytotoxicity (**D**). The data are expressed as mean values ± the SE of two separate experiments. Cells were isolated from three independent donors and assayed in triplicate.

**Table 1 molecules-25-00842-t001:** Mass spectrometric conditions and contents of the detected compounds, their retention times (Rt), and their molecular ([M − H]^−^) and fragment ions in negative and positive mode in *Chelidonium majus* root extract.

No	Compound	Retention Time (min)	Parent Ion (*m/z*)	Product Ion (*m/z*)	Ion Mode	Content (µg/g)
1.	malic acid	1.08	133.1	115, 71	−	p
2.	*trans*-aconitic acid	1.15	172.9	85, 129	−	p
3.	quinic acid	1.3	191	85, 93	−	p
4.	protocatechuic acid	1.54	153	109, 108	−	LOQ
5.	vanillic acid	2.37	167.3	151.9, 107.9	−	LOD
6.	*trans*-caffeic acid	2.71	179.2	135, 134, 89	−	p
7.	salicylic acid	2.79	137.3	93, 65.05, 44.8	−	LOD
8.	hydroxybenzoic acid	2.85	137.3	93	−	LOQ
9.	*p*-coumaric acid	3.97	163.2	119.1, 93.1, 117	−	p
10.	rosmarinic acid	4.98	359.1	161, 197.15	−	LOD
11.	vanillin	5.15	151.2	136, 91.85, 108	−	p
12.	protopine derivative	6.75	354	320, 260, 196	+	p
13.	allocryptopine	6.80	369.6	352, 187.95, 290	+	98.94 ± 0.02
14.	quinine sulfate	7.00	747.4	325.1	+	p
15.	coptisine	7.50	320.1	291.95, 204.05, 262.05	+	1077.06 ± 26.24
16.	berberine	7.95	336.4	320, 292, 321.1	+	722.62 ± 19.25
17.	quercetin	8.34	301.1	151.05, 65, 121	−	LOD
18.	chelidonine derivative	8.84	370	356, 339	+	p
19.	chelidonine	9.56	353.8	275, 189, 247	+	1181.41 ± 72.78
20.	chelerythrine	10.44	348.1	332, 304, 333	+	1761.22 ± 33.80
21.	tetrahydroberberine	10.55	340	176, 149	+	p
22.	tetrahydrocoptisine	11.10	324	176, 149	+	p
23.	coptisine derivative	11.20	324	190	+	p
24.	sanguinarine	11.80	332.1	274.1, 316.95, 246	+	1373.80 ± 27.50
25.	protopine	12.76	320.2	303.2, 107, 123.85	*+*	p

Here, p—present, where identification was based on mass spectra with no reference substances; LOD—limit of detection; LOQ—limit of quantification.

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
