# Peer review of "Modulatory Effect of *Chelidonium majus* Extract and Its Alkaloids on LPS-Stimulated Cytokine Secretion in Human Neutrophils"

_molecules, 2020, doi:10.3390/molecules25040842_

Round 1
Reviewer 1 Report
The authors have examined the crude extract and the major alkaloids of Chelidonium majus on the secretion of IL-1β, IL-8, and TNF-α in human neutrophils.
Why weren't the non-alkaloid components (e.g., p-coumaric acid, salicylic acid, vanilic acid, caffeic acid, vanillin, rosmarinic acid, or quercetin) screened for bioactivity? Were the activities too low to be considered?
Some minor editorial corrections:
The scientific names in the Abstract need to be italicized.
Line 183: “…proportions (Figure 2, Table 1), which(?) could explain…”
Line 222: benzylphthalides(?)
Reviewer 2 Report
The authors have done a preliminary study for the anti-inflammatory effect of the crude extract of Chelidonium majus and its five major alkaloids. They have a good experimental design for both the chemical analysis by using LC-MRM-MS quantification strategy and multiple complementary bioassays to evaluate the bioactivity of crude extract and selected alkaloids. It was revealed that the crude extract showed no cytotoxicity at lower concentration but can’t decrease none of the three cytokines level neither. Although most of individual alkaloid has shown no cytotoxicity except coptisine and sanguinarine, only sanguinarine could decreased the cytokine level significantly at the concentration of 2.5 uM at which it became toxic. Based on our previous experience, I think there must be some potential bioactive ingredients with much lower abundance present in the extract but other than alkaloids. I suggest the authors to re-screen the bioactivity of crude extract at several more different concentrations lower than 6.25 ug/mL in order to get a concentration-dependent manner. Then proceed to target the potential bioactive ingredients by applying mass spectrometry-based metabolomics strategy integrated with multivariate analysis. The authors have made a good effort but I feel sorry to say their study is not qualified to be published at current stage.
Reviewer 3 Report
The present manuscript describes the characterization of isoquinoline alkaloids from Chelidonium majus along with biological activities. In particular it has been demonstrated that some of the phenanthridine and protoberberine derivatives as well as Chelidonium majus extract present anti-inflammatory and cytotoxic properties. From an analytical viewpoint liquid chromatography coupled to tandem mass spectrometry was employed for the determination of alkaloids, phenolic acids, carboxylic acids and hydroxybenzoic acids and a total of
25 different compounds were positively identified. The work has been conducted with adequate means. The paper is well-written and well-organized. The most significant finding is that isoquinoline alkaloids were found to be active at very low concentration levels.
Reviewer 4 Report
Dear Authors
My comments and suggestions are in the attached file.
Best regards

Round 2
Reviewer 2 Report
The authors has argue a lot for their results but not their experimental conditions which I think they should take more consideration and spend more effort to improve:
First, the five alkaloids they selected are commonly existed no matter naturally or synthesized and their anti-inflammatory effect in vitro (and in-vivo) have been well studied. All of them have shown anti-inflammatory activities in concentration-dependent manner ( Reference 1-5). In this manuscript, the authors try to stitch these five well-studied compound with Chelidonium majus extract as a story but their experimental results did not support their hypothesis because the extract showed significant toxicity even at the lowest test concentration of 1.25 ug/mL. This leads to the failure to trace any bioactive ingredient can exhibit sound anti-inflammatory activity even they have tried three different cytokines models. LPS has been used as a toxin to induce inflammation for the cell in the three bioassays in this manuscript. As to show anti-inflammatory activity, the test compounds/extract should reduce the inflammation more as the concentration increase (basic knowledge for biologists), but there is no trend can be found from their results (Figure 3, 4 and 5) to show the activities. How this data can be published if there is no evidence to support their conclusion? The authors should think carefully about their experimental design to obtain a solid and reliable result fulfilled with basic norm of science. I suggest the authors to read the references carefully to re-design their experimental conditions to get the data qualified to be published in future.References:
1. coptisine (Cmin=1 uM)
https://www.ncbi.nlm.nih.gov/pubmed/27018392
2. berberine (Cmin=0.1 uM)
https://link.springer.com/article/10.1007/s10753-010-9276-2
3.chelidonine (Cmin=5 uM)
https://www.ncbi.nlm.nih.gov/pubmed/30257328
4.chelerythrine (Cmin=0.5 uM)
https://www.frontiersin.org/articles/10.3389/fphar.2018.01047/full
5.Sanguinarine (Cmin=0.5 uM)
https://link.springer.com/article/10.1007%2Fs11596-018-1867-4